# Sentinel Lymph Node Mapping by Retroperitoneal vNOTES for Uterus-Confined Malignancies: A Standardized 10-Step Approach

**DOI:** 10.3390/cancers16112142

**Published:** 2024-06-05

**Authors:** Daniela Huber, Yannick Hurni

**Affiliations:** 1Department of Gynecology and Obstetrics, Valais Hospital, Av. Du Grand-Champsec 80, 1951 Sion, Switzerland; 2Department of Pediatrics, Gynecology and Obstetrics, Geneva University Hospitals, Bd. de la Cluse 30, 1205 Geneva, Switzerland; yannick.hurni@hopitalvs.ch

**Keywords:** endometrial cancer, cervical cancer, sentinel lymph node, vNOTES, transvaginal natural orifice transluminal endoscopic surgery, minimally invasive surgery

## Abstract

**Simple Summary:**

Sentinel lymph node (SLN) mapping for the surgical staging of endometrial and cervical cancer is commonly performed by laparoscopy, but recently, a new retroperitoneal transvaginal natural orifice transluminal endoscopic surgery approach has been described and developed by Jan Baekelandt. This technique provides easy visualization of lymphatic afferent vessels and pelvic lymph nodes, early SLN assessment, and a coherent mapping methodology following the lymphatic flow from caudal to cranial. However, only a few publications have reported it. Following the IDEAL (Idea Development Exploration Assessment Long-term follow-up) framework, research concerning this technique is in Stage 2a, with only small case series as evidence of its feasibility. Provided that the standardized description of this surgical technique appears necessary to provide the homogeneity required to move further in its investigation, here, we describe a 10-step approach for performing it successfully. This could help other surgeons approach this new technique, and it proposes a common methodology necessary for evolving through future studies.

**Abstract:**

(1) Background: Sentinel lymph node (SLN) mapping represents an accurate and feasible technique for the surgical staging of endometrial and cervical cancer. This is commonly performed by conventional laparoscopy or robotic-assisted laparoscopy, but in recent years, a new retroperitoneal transvaginal natural orifice transluminal endoscopic surgery (vNOTES) approach has been described and developed by Jan Baekelandt. This technique provides easy visualization of lymphatic afferent vessels and pelvic lymph nodes, early SLN assessment, and a coherent mapping methodology following the lymphatic flow from caudal to cranial. However, only a few publications have reported it. Following the IDEAL (Idea Development Exploration Assessment Long-term follow-up) framework, research concerning this technique is in Stage 2a, with only small case series as evidence of its feasibility. Its standardized description appears necessary to provide the surgical homogeneity required to move further. (2) Methods: Description of a standardized approach for retroperitoneal pelvic SLN mapping by vNOTES. (3) Results: We describe a 10-step approach to successfully perform retroperitoneal vNOTES SLN mapping, including pre-, intra-, and postoperative management. (4) Conclusions: This IDEAL Stage 2a study could help other surgeons approach this new technique, and it proposes a common methodology necessary for evolving through future IDEAL Stage 2b (multi-center studies) and Stage 3 (randomized controlled trials) studies.

## 1. Introduction

Surgical staging of lymph nodes defines the stage and the treatment plan for uterus-confined malignancies [1,2]. In recent years, complete regional lymphadenectomy, associated with increased morbidity and occasionally severe postoperative complications, was replaced by targeted lymph node sampling in a significant proportion of patients with early-stage diseases [3]. Sentinel lymph node (SLN) mapping represents an accurate and feasible technique for the surgical staging of early-stage endometrial and cervical cancer [1,2,3,4,5,6,7,8]. In addition, it allows for the identification of unexpected drainage pathways and low-volume nodal disease [9]. SLN mapping is commonly performed by conventional laparoscopy (CL) or robotic-assisted laparoscopy [3,10,11], but in recent years, new transvaginal natural orifice transluminal endoscopic surgery (vNOTES) approaches, both transperitoneal and retroperitoneal, have been described [12,13]. The retroperitoneal vNOTES approach appears to be the most promising, providing easy visualization of lymphatic afferent vessels and pelvic lymph nodes (LNs), early LN assessment, and a coherent mapping methodology following the lymphatic flow from caudal to cranial [14]. Although this technique appears to be a possible revolutionary tool to stage endometrial and cervical cancer, only a few publications have reported it [12,14,15,16,17,18,19].

Following the IDEAL (Idea Development Exploration Assessment Long-term follow-up) framework, research concerning this technique is in Stage 2a, with only small case series as evidence of its feasibility. For this reason, its standardized description appears necessary to provide the surgical homogeneity required to move to larger multi-center studies and, subsequently, randomized controlled studies to compare it with the laparoscopic techniques. 

In this study, we aim to describe a standardized approach for retroperitoneal pelvic SLN mapping by vNOTES, including pre-, intra-, and postoperative management. 

## 2. Materials and Methods

Since October 2021, retroperitoneal vNOTES SLN mapping has been used in our institution for the surgical staging of more than 45 patients with early-stage endometrial or cervical cancer. Based on our experience, below, we describe a standardized step-by-step approach to successfully performing this technique. A video tutorial (Appendix A) is available in the online Appendix A.

## 3. Results

### 3.1. Step-by-Step Surgical Technique

#### 3.1.1. Step One—Preparation and Positioning

Patients should receive prophylactic antibiotics according to the hospital’s policy. We suggest administering a single dose of clindamycin vaginal cream 2% (5 g of cream with 100 mg of clindamycin) the day before surgery and 2–4 h before intervention, in addition to cefuroxime 1.5 g (3 g for patients weighing more than 80 Kg) and metronidazole 500 mg intravenously upon the induction of anesthesia [20,21]. A repeat dose of 1.5 g of cefuroxime must be provided if the intervention lasts ≥ 3 h. Under general anesthesia and muscular relaxation, patients are positioned in a horizontal dorsal lithotomy position, and a bladder catheter is placed.

#### 3.1.2. Step Two—Cervical Injection of Indocyanine Green

Vaginal retractors are placed in the vagina, and the cervix is grasped with a Pozzi forceps at the 12 o’clock position. Indocyanine green (ICG) is injected superficially (1–2 mm) into the cervix at the 3 and 9 o’clock positions, with a total of 2 to 4 mL solution at a 1.25–2.5 mg/mL concentration. This injection can be repeated in the absence of ICG-positive LNs. 

#### 3.1.3. Step Three—Colpotomy and Access to the Pelvic Retroperitoneal Space

The pelvic retroperitoneal space can be reached through a single midline incision into the anterior vaginal wall [22] or two separate incisions into the lateral vaginal fornices [12]. 

For the single-incision approach, the anterior vaginal wall is grasped with two Allis forceps placed on the midline close to the cervix. Approximately 10 mL of a mixture prepared with 20 mL of ropivacaine 0.5% and 20 mL of epinephrine 0.1% are infiltrated under the mucosa of the anterior vaginal wall for hydrodissection and vasoconstriction to enhance blunt dissection and reduce bleeding. A 4 cm midline incision is made between the Allis forceps with a scalpel. The margins of the incision are grasped with Allis forceps, and the vaginal mucosa is dissected from the bladder, initially using scissors and then with blunt dissection using a gauze-covered finger. The dissection is directed laterally at approximately 45° on a horizontal plane, trying to stay as close as possible to the vaginal wall and to the pelvic wall to reduce the risk of bladder injuries. This dissection allows for opening the firm visceral endopelvic fascia to reach the looser pelvic retroperitoneal space.

For the bilateral-incision approach, the lateral vaginal wall is grasped with two Allis forceps placed at 8 and 10 o’clock positions on the right side (2 and 4 o’clock positions on the left side) into the lateral vaginal fornix. Approximately 5 mL of the same mixture described previously is injected under the mucosa of the lateral vaginal wall. A 3 cm incision is created between the Allis forceps with a scalpel. The margins of the incision are grasped with Allis forceps, and lateral dissection is performed with the alternate use of scissors and a gauze-covered finger, developing the paravesical space. This dissection is very similar to that described for the single-incision technique.

The same procedure is performed contralaterally.

#### 3.1.4. Step Four—vNOTES Port Installation

A GelPoint V-Path Transvaginal Access Platform (Applied Medical, Rancho Santa Margarita, CA, USA) with an Alexis retractor inner ring with a 7 cm diameter is used as a vNOTES port. The Alexis retractor inner ring is inserted into the right or left paravesical space, 2–3 cm deeper than the vaginal mucosa, using its introducer. Using Doyen or Breisky retractors inserted into the paravesical space can facilitate the Alexis retractor positioning. Minimal traction should be exerted on the Alexis retractor (the plastic sleeve is rolled over the outer ring 1 or 2 times) to reduce the risk of dislocation. Three 10-mm trocars are placed into the GelSeal Cap at 4, 6, and 8 o’clock positions. 

#### 3.1.5. Step Five—Endoscopic Pelvic Retroperitoneal Space Preparation

Once the vNOTES port is installed, carbon dioxide is insufflated to expand the retroperitoneal space to a 10–15 mmHg pressure. Blunt dissection using an atraumatic grasper or a suction cannula is initially performed in medial and ventral directions, carefully pushing the bladder and the parietal peritoneum to enlarge the pelvic retroperitoneal space. This dissection is continued to identify the pelvic anatomical structures (Figure 1 and Figure 2). We suggest that the obturator nerve and the external iliac artery and vein must be identified before any lymph nodal dissection. The ureter; the obturator vessels; and the umbilical, internal iliac, and common iliac arteries should be optionally identified depending on the localization of SLNs. The presacral region should only be prepared in the case of negative SLN mapping into other pelvic regions. 

#### 3.1.6. Step Six—Sentinel Lymph Node Identification

The pelvic retroperitoneal space is inspected for ICG uptake by LNs using a near-infrared fluorescent optic device connected to a 30° scope. Inspection starts by identifying the ICG-positive afferent lymph vessels from the cervix or uterus, and following their path toward one or more pelvic lymphatic stations in the obturator, external iliac (Figure 3), internal iliac, common iliac, or presacral regions. SLN is the most proximal ICG-positive LN with a clear afferent vessel from the cervix or uterus. Multiple ICG-positive nodes from the same lymphatic group should all be deemed as SLNs and biopsied. Multiple ICG-positive nodes from different groups should be considered SLNs only if specific and separate afferent lymphatic vessels coming from the cervix or uterus are demonstrated. In contrast, second-echelon ICG-positive nodes should not be harvested systematically. 

We previously described this retroperitoneal vNOTES approach for SLN mapping and demonstrated it in a video, Available online: https://www.jmig.org/article/S1553-4650(24)00080-3/fulltext#supplementaryMaterial (accessed on 3 June 2024) [14]. 

#### 3.1.7. Step Seven—Sentinel Lymph Node Harvesting

Once identified, SLNs are separated from the surrounding tissues by careful dissection using a bipolar sealing device (Figure 4). Afferent and efferent lymphatic channels should be identified and adequately coagulated before their section. SLNs are extracted transvaginally without requiring extraction bags, just removing the GelSeal Cap to avoid smashing the LNs. In addition to ICG-fluorescence-positive SLNs, any other suspicious lymph nodes should be harvested.

#### 3.1.8. Step Eight—Vaginal Suture

After meticulous hemostasis using a bipolar grasper, the vNOTES port is extracted. The vaginal mucosa is then closed with simple stitches using Vicryl 0 both for the midline and lateral incisions. No drainage tubes are needed. Compression with gauze packed into the vagina for 4–24 h can be performed to reduce the risk of pelvic retroperitoneal hematoma. 

#### 3.1.9. Step Nine—Additional Interventions

To complete the staging of early-stage endometrial cancer, a simple hysterectomy with bilateral salpingectomy or salpingo-oophorectomy can then be performed by conventional vaginal surgery, vNOTES [15], or CL (rarely by laparotomy), depending on the situation. In the case of vNOTES hysterectomy, we suggest suturing the cervical canal after the injection of ICG to reduce the risk of intrabdominal or retroperitoneal cancerous cell spillage. The vNOTES approach can also be used to perform infracolic omentectomies needed for the surgical staging of high-risk endometrial cancers [23,24]. In the case of early-stage cervical cancer, SLN mapping can be associated with a simple or radical trachelectomy by conventional vaginal surgery [25] or with a radical hysterectomy by vNOTES [26]. 

#### 3.1.10. Step Ten—Postoperative Care

Patients should receive a single dose of clindamycin vaginal cream 2% (5 g of cream with 100 mg of clindamycin) once a day during the first 7 postoperative days. In the case of vaginal packing, we recommend maintaining the bladder catheter to reduce the risk of urinary retention. Depending on the situation, patients can be hospitalized for 1–2 postoperative nights, or, occasionally, interventions in a daycare setting can be proposed. 

## 4. Discussion

We proposed a standardized 10-step approach for SLN mapping by retroperitoneal vNOTES for patients with early-stage endometrial and cervical cancer. Since Baekelandt first described this technique in 2019 [12], only a few articles have been published on this topic, including case reports, small single-center case series, and proposals for technique modifications [14,15,16,17,18,19,22]. Following the IDEAL (Idea Development Exploration Assessment Long-term follow-up) framework, we can consider that research regarding this technique is in Stage 2a (Development), and its standardized description appeared necessary before moving to Stage 2b (Exploration), allowing for a greater surgical homogeneity required to conduct larger multi-center studies. Below, we discuss some specific aspects that must be understood in order to correctly perform this new surgical approach to SLN mapping, in addition to a clarification regarding its advantages and weaknesses. 

During a retroperitoneal vNOTES SLN mapping, patients do not need to be placed in a Trendelenburg position as for a CL approach. This could be an interesting advantage in patients who do not tolerate steep Trendelenburg positions, such as obese women [27,28].

We decided to routinely place a bladder catheter during retroperitoneal vNOTES SLN mapping to reduce the risk of bladder injury and allow for a better expansion of the pelvic paravesical retroperitoneal space. However, some surgeons prefer to keep the bladder moderately filled during the intervention to quickly recognize a possible bladder injury, especially during the vesicovaginal dissection needed to access the retroperitoneal space through a midline single-incision approach. 

This surgical technique was initially described with pelvic retroperitoneal accesses through two separate incisions on the lateral walls of the vagina [12]. Subsequently, a new anterior vaginal wall midline single-incision approach was described [22]. The latter presents the advantage of a more rapid single-incision approach to reach both paravesical spaces. In addition, it could be easier to learn for gynecological surgeons, as the initial vesicovaginal dissection is similar to that performed for an anterior colporrhaphy and the deeper dissection is similar to that needed to perform a tension-free vaginal tape obturator (TVT-O) or a transobturator tape (TOT). Because of the proximity to the urethra, we suggest performing this dissection close to the cervix to reduce the risk of urethral hypermobility and iatrogenic postoperative stress urinary incontinence. In addition, the proximity to the bladder increases the risk of iatrogenic injury to this organ. For this reason, we suggest performing the anterior access for patients presenting at least a small cystocele, which makes dissection easier. Conversely, in the case of deep and narrow vagina, we suggest using the lateral approach to reduce the risk of bladder injury. 

Both approaches can lead to a successful retroperitoneal SLN mapping if correctly performed. 

Bladder injury during paravesical space is the most dreaded complication. This typically involves the lateral bladder wall near the trigone. These injuries are often immediately recognized, easily repaired transvaginally, and do not prevent completion of the SLN mapping or associated interventions. In the case of bladder injury, we suggest immediately repairing it to avoid the risk of inadvertent intravesical insufflation of carbon dioxide in the continuation of the intervention. In addition, we suggest routinely performing an intraoperative cystoscopy to evaluate the bladder’s inside wall and the ureteral patency. 

Once the pelvic retroperitoneal space has been opened, the inner ring of the Alexis retractor should be inserted into the obturator fossa, enough to stay in place but not too much to avoid covering the distal part of the obturator, external iliac, and internal iliac lymphatic regions. In addition, minimal traction should be exerted to prevent accidental displacement of the retractor. 

According to international consensuses regarding the standardization of the laparoscopic techniques in SLN mapping for endometrial and cervical cancer, the identification of some anatomic structures, such as the external and internal iliac vessels, the umbilical artery, and the ureter, should be mandatory before performing LN dissections [29,30]. In addition, SLN mapping should start at the level of the uterine artery and continue laterally away from the uterus [29]. However, these recommendations do not appear to be relevant to the retroperitoneal vNOTES approach. In this case, the dissection is performed from caudal to cranial, following the afferent lymphatic vessels in a physiological way from the uterus toward the pelvic lymphatic stations. This allows for the immediate identification of the area where SLNs are located, and the dissection can then be directed more specifically in this direction: laterally for the obturator or external iliac regions or medially for the internal iliac, common iliac, or presacral regions. Considering that up to 90% of the SLNs are encountered in the lateral regions (obturator or external iliac regions) [11,14,31,32], we suggest that the obturator nerve and the external iliac vessels should be the primary structures to identify. The internal iliac and common iliac vessels, the umbilical and uterine arteries, and the ureter should be identified in the case of afferent ICG-positive lymphatic vessels with a clear path along the medial part of the pelvic retroperitoneal space. As opposed to transperitoneal techniques, this spatial distinction between lateral and medial regions is accentuated through this vNOTES retroperitoneal approach, with the pressurized carbon dioxide insufflation allowing for clear separation of the lateral from the medial sensitive structures. In addition, the direction of the dissection from caudal to cranial allows for, in most cases, encountering SLNs nearer than the sensitive pelvic anatomical structures that lie further away, in contrast with the transabdominal approaches. This makes it unnecessary to identify all of the medial structures in the case of an obturator or external iliac SLN, provided that these are located at a safe distance. Para-aortic and presacral SLNs are rare, and we do not suggest routinely screening these regions. However, if necessary, these also appear accessible via a retroperitoneal vNOTES approach [12,33]. This minimal approach decreases the operating time and reduces the risk of injury to anatomical structures that are located at a safe distance from the lymphatic vessels and SLNs of interest. 

Another difference between the CL and the retroperitoneal vNOTES approach concerns how to extract SLNs. While using an extractor device appears mandatory in a CL approach [29,30], this is not the case with retroperitoneal vNOTES, provided that the space through the Alexis retractor is enough to avoid smashing the LNs. However, we suggest removing the GelSeal Cap to remove the LNs without passing them through the trocars. 

The retroperitoneal vNOTES approach for SLN mapping could be part of the complete surgical management of early-stage endometrial cancer by vNOTES, which can be associated with total hysterectomy, salpingo-oophorectomy, and omentectomy [14,15,23]. In the context of early-stage cervical cancer diagnosed on conization, this vNOTES approach represents a valuable option in a two-step strategy with initial SLN mapping with definitive pathological analyses, followed by a radical hysterectomy [1,19,26] or a simple or radical trachelectomy if LNs appear negative [25]. Recent randomized data support that parametrial infiltration is very low in early-stage and low-risk cervical cancer [34]. Another recent prospective trial showed that conservative surgery with conization or simple hysterectomy is feasible in early stages and low-risk cervical cancer [35]. The patients were staged mainly by classical laparoscopy with 5% positive nodes and 2.5% residual disease in postconisation hysterectomy specimens. In this setting, where conization or simple hysterectomy with lymph node staging is a safe oncological option, vNOTES retroperitoneal sentinel node biopsy may offer a new vaginal and completely extraperitoneal option to the classical laparoscopic approach. 

This vNOTES technique has several potential advantages over laparoscopic approaches, such as sentinel dissection without Trendelenburg positioning; a caudal to cranial LNs inspection following the natural lymphatic distribution upwards, which could improve the identification of true SLNs and not secondary or higher nodes situated on the sentinel pathway; better access to LNs situated under the external iliac vein; and a less invasive approach avoiding transabdominal incisions with reduced risks of adhesion formation potentially responsible for the severe side effects associated with postoperative radiotherapy in the case of cervical cancer with LN involvement [1,36]. In addition, vNOTES approaches could reduce operative times, present reduced postoperative pain, and decrease hospital stays [37,38,39].

Retroperitoneal vNOTES SLN mapping presents some limitations, mainly associated with difficulty in accessing the pelvic retroperitoneal space in patients with a deep and narrow vagina. In these cases, both lateral and anterior accesses could be very difficult, and surgical staging by CL may sometimes be indicated. Other limitations are reduced instrument triangulation and restricted anatomical spaces, but using articulating instruments can help overcome these constraints. 

## 5. Conclusions

In this IDEAL Stage 2a study, we propose a standardized approach to retroperitoneal vNOTES SLN mapping for patients with early-stage endometrial and cervical cancer. This could help other surgeons approach this new technique, and it proposes a common methodology necessary for evolving through future IDEAL Stage 2b (multi-center observational studies) and Stage 3 (randomized controlled trials vs. CL) studies. 

## Figures and Tables

**Figure 1 cancers-16-02142-f001:**
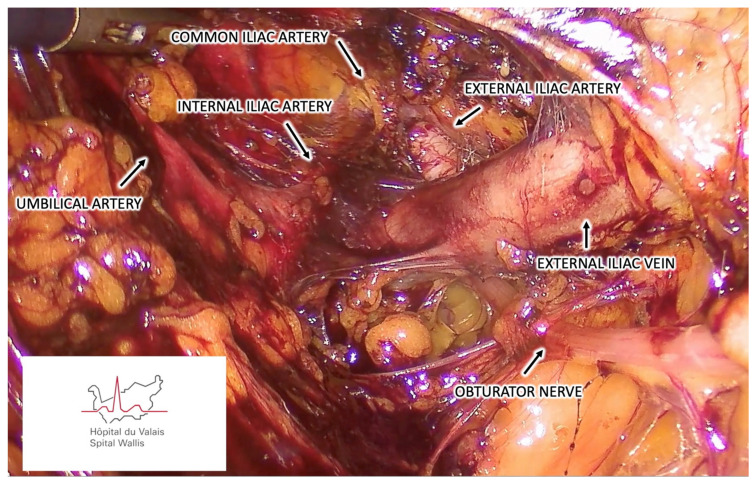
Retroperitoneal vNOTES view demonstrating the left pelvic anatomical structures.

**Figure 2 cancers-16-02142-f002:**
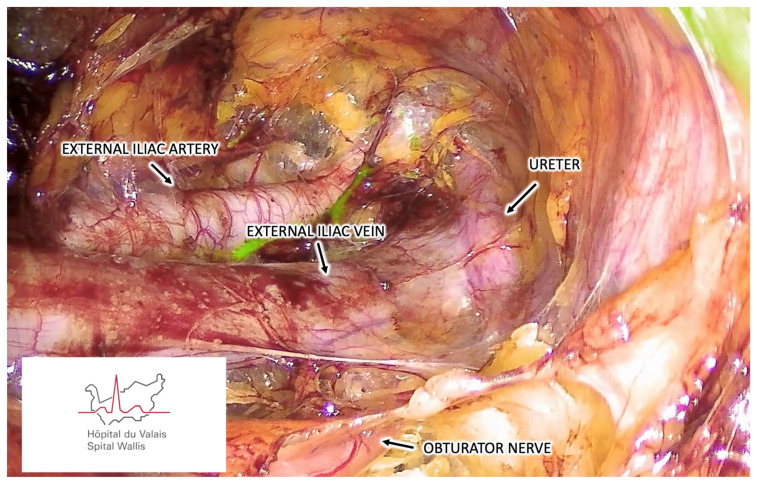
Retroperitoneal vNOTES view demonstrating the right pelvic anatomical structures.

**Figure 3 cancers-16-02142-f003:**
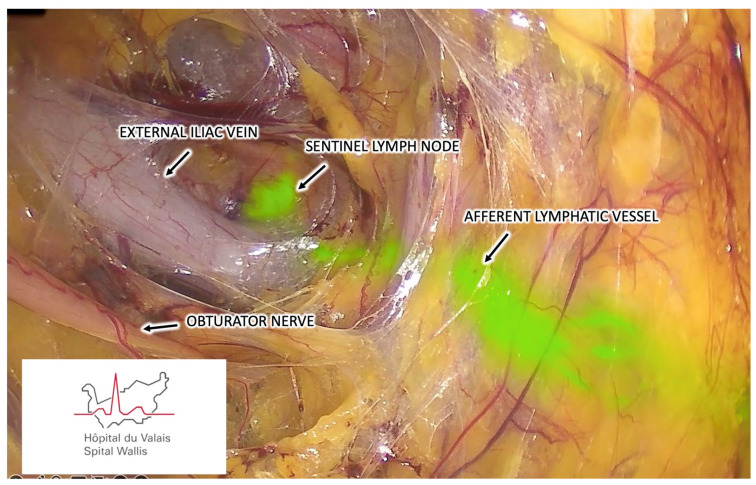
Retroperitoneal vNOTES view demonstrating SLN mapping to the right external iliac region.

**Figure 4 cancers-16-02142-f004:**
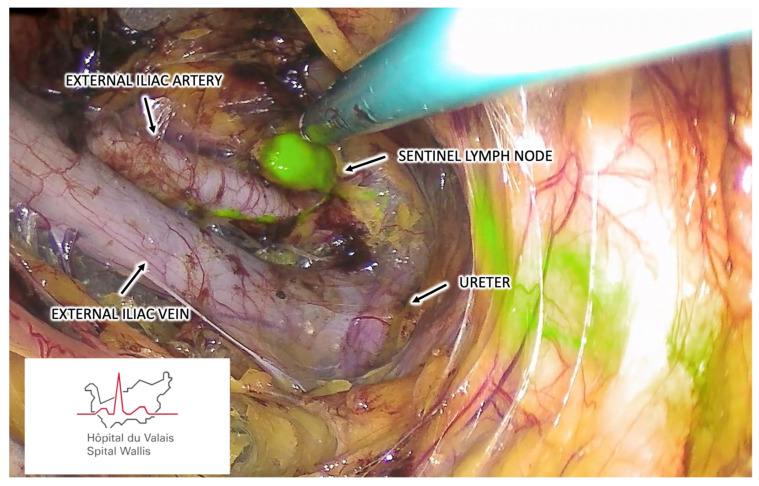
Demonstration of an SLN harvested by retroperitoneal vNOTES from the right external iliac region.

## Data Availability

No new data were created or analyzed in this study. Data sharing is not applicable to this article.

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
