# Peer review of "Sentinel Lymph Node Mapping by Retroperitoneal vNOTES for Uterus-Confined Malignancies: A Standardized 10-Step Approach"

_cancers, 2024, doi:10.3390/cancers16112142_

Round 1
Reviewer 1 Report
Comments and Suggestions for Authors
The article presents a fairly detailed description of the steps to follow to access the retroperitoneum via the vagina V-Notes.
Before accepting the article for publication, I would like to suggest the inclusion of a video that shows how to perform the approach
In the discussion section, the authors should include an analysis of the possible oncological outcomes of this vaginal access, especially when the text refers to the possibility of using V-Notes in early cervical cancer, ignoring the data from the LACC study. The latter seems to me to be essential to avoid possible future problems
Author Response
Sion, May 22nd 2024
Dear Editor and Reviewer,
Thank you for giving us the opportunity to submit a revised draft of our manuscript titled
“Sentinel Lymph Node Mapping by Retroperitoneal vNOTES for Uterus-Confined Malignancies: A Standardized 10-Step Approach” to Cancers. We appreciate the time and effort that you and the reviewers have dedicated to providing your valuable feedback on our manuscript. We are grateful to the reviewers for their insightful comments on our paper. We have been able to incorporate changes to reflect most of the suggestions provided by the reviewers. We have highlighted the changes within the manuscript.
Here is a point-by-point response to the reviewers’ comments and concerns.
REVIEWER 1, POINT 1
- Before accepting the article for publication, I would like to suggest the inclusion of a video that shows how to perform the approach.
We agree with your suggestion, therefore, we added the requested video of retroperitoneal sentinel biopsy by vNOTES approach.
REVIEWER 1, POINT 2
- In the discussion section, the authors should include an analysis of the possible oncological outcomes of this vaginal access, especially when the text refers to the possibility of using V-Notes in early cervical cancer, ignoring the data from the LACC study. The latter seems to me to be essential to avoid possible future problems
- Are the conclusions supported by the results? Reviewer: Must be improved.
Thank you for pointing this out. We agree with your suggestion and we have added a paragraph to emphasize this topic (paragraph 4, page 9).
REVIEWER 1
- Does the introduction provide sufficient background and include all relevant references? Reviewer : Must be improved
We agree with your suggestion. We added more data about the impact of surgical lymph node staging in uterine malignancies in the introduction section (paragraph 1 page 2).
Reviewer 2 Report
Comments and Suggestions for Authors
First of all, thank you to the authors for such a detailed and well-conducted work.
Improvements in approach and detection procedures for staging have been progressively achieved by minimally invasive techniques.
As the authors have well defined and cited, they are small initial series with case descriptions. Here we find a very detailed contribution, and the idea of the 10-step approach proves to be quite useful.
It comes from authors who have published and been part of the establishment of the use (referenced in 5 of the publications cited).
I congratulate the authors for the standardization that shows their expertise and only contributes to the next expected phase (Exploration).
Author Response
Sion, May 22nd 2024
Dear Editor and Reviewer,
Thank you for giving us the opportunity to submit a revised draft of our manuscript titled
“Sentinel Lymph Node Mapping by Retroperitoneal vNOTES for Uterus-Confined Malignancies: A Standardized 10-Step Approach” to Cancers. We appreciate the time and effort that you and the reviewers have dedicated to providing your valuable feedback on our manuscript. We are grateful to the reviewers for their insightful comments on our paper. We have been able to incorporate changes to reflect most of the suggestions provided by the reviewers. We have highlighted the changes within the manuscript.
Here is a point-by-point response to the reviewers’ comments and concerns.
REVIEWER 2
- Thank you for your comments and kind review.
Reviewer 3 Report
Comments and Suggestions for Authors
The manuscript submitted for review concerns an interesting and important problem of creating standards for an application of a new technique for identifying and harvesting the sentinel node in patients with malignant tumors of the genital organs.
Creating gold standards is very important because it allows to become familiar with new technology, allows to apply it correctly, repeat it, and allows to compare the results obtained in a standardized way. This publication may obtain many citations. The only question remains: Who should create such standards? I believe that the group of 45 operations is rather small to be used to set universal, global standards, but in the light of the lack of other, more numerous observations, this is where we should start. The authors themselves define the level of research on the method as "Development".
The description of the consecutive steps is presented in detail and clearly. The discussion is written in a very good way, it is clear, and factual and describes the historical background of the method interestingly.
I believe that the manuscript should be published and will constitute a valuable reference point for subsequent publications.
Author Response
Dear Editor and Reviewer,
Thank you for giving us the opportunity to submit a revised draft of our manuscript titled
“Sentinel Lymph Node Mapping by Retroperitoneal vNOTES for Uterus-Confined Malignancies: A Standardized 10-Step Approach” to Cancers. We appreciate the time and effort that you and the reviewers have dedicated to providing your valuable feedback on our manuscript. We are grateful to the reviewers for their insightful comments on our paper. We have been able to incorporate changes to reflect most of the suggestions provided by the reviewers. We have highlighted the changes within the manuscript.
Here is a point-by-point response to the reviewers’ comments and concerns.
REVIEWER 3
- Thank you for your comments and kind review.